# Exploring the Limits of Domain-Adaptive Training for Detoxifying Large-Scale Language Models

**Boxin Wang**[*][†][1], **Wei Ping**[†][2], **Chaowei Xiao**[†][2,3], **Peng Xu**[2], **Mostofa Patwary**[2],
**Mohammad Shoeybi**[2], **Bo Li**[1], **Anima Anandkumar**[2,4], and **Bryan Catanzaro**[2]

[1]University of Illinois at Urbana-Champaign
[2]NVIDIA   [3]Arizona State University  [4]California Institute of Technology

## Abstract

Pre-trained language models (LMs) are shown to easily generate toxic language. In this work, we systematically explore domain-adaptive training to reduce the toxicity of language models. We conduct this study on three dimensions: training corpus, model size, and parameter efficiency. For the training corpus, we demonstrate that using self-generated datasets consistently outperforms the existing baselines across various model sizes on both automatic and human evaluations, even when it uses a $\frac{1}{3}$ smaller training corpus. We then comprehensively study detoxifying LMs with parameter sizes ranging from 126M up to 530B ($3\times$ larger than GPT-3), a scale that has never been studied before. We find that *i)* large LMs have similar toxicity levels as smaller ones given the same pre-training corpus, and *ii)* large LMs require more endeavor to unlearn the toxic content seen at pre-training. We also explore parameter-efficient training methods for detoxification. We demonstrate that adding and training *adapter*-only layers in LMs not only saves a lot of parameters but also achieves a better trade-off between toxicity and perplexity than whole model adaptation for large-scale models. Our code will be available at: https://github.com/NVIDIA/Megatron-LM/.

## 1   Introduction

Large-scale pre-trained language models (LMs) [1–6] have demonstrated substantial performance gains on various NLP tasks, especially when scaling up the sizes of models. However, recent studies [7, 8] show that generative LMs can generate toxic and biased language, which raises ethical concerns for their safe deployment in real-world applications.

Previous methods on reducing the toxicity of LMs can be categorized as: *decoding-time* methods, *pre-training-based* methods, and *domain-adaptive training* methods. Decoding-time methods [9–14] manipulate the output distribution or input prompts at the inference stage without modifying the original model parameters. These methods can be flexible, but they either resort to some simple word filtering strategies [10], or increase the computational cost at the inference stage. For example, PPLM [9] requires multiple iterations of backward propagation through the LM when generating every token, which makes it prohibitively expensive to be deployed to production especially for large-scale LMs. [3] In contrast, *pre-training-based* methods directly filter out the potentially toxic

---

[*]Work done during an internship at NVIDIA.

[†]Correspondence to: Boxin Wang <boxinw2@illinois.edu>, Wei Ping <wping@nvidia.com>, Chaowei Xiao <xiaocw@asu.edu>.

[3]For example, the 530B Megatron-Turing NLG [6] requires 16 A100 80GB GPUs for autoregressive generation, but 280 GPUs for backward propagation for memory reasons.

36th Conference on Neural Information Processing Systems (NeurIPS 2022).

content within the pre-training corpus and retrain the model from scratch [e.g., 15]. However, it is difficult to determine the filtering criterion beforehand, and pre-training a large LM multiple times from scratch is quite expensive.

Domain-adaptive training methods [10, 16] further fine-tune the pre-trained LMs on carefully curated datasets (e.g., Jigsaw, filtered OWTC [17]). For instance, Gehman et al. [10] construct a nontoxic data corpus from an existing dataset, OWTC, via the Perspective API [4] and perform the fine-tuning on the nontoxic corpus. Domain-adaptive training is more flexible than pre-training methods, as one can still customize the model after the expensive pre-training process. Compared to the decoding-time methods, domain-adaptive training methods have the following advantages: *i*) they can achieve fast and memory-efficient inference, thus can be deployed in broader systems; and *ii*) they can largely reduce the model toxicity while still maintaining good LM quality measured by perplexity and downstream task performance as we will show in this work.

In this paper, we explore the limits of domain-adaptive training for detoxifying language models along the following three aspects: *1*) *Training Corpus*: Unlike previous methods using curated pre-training corpus for detoxification, we propose to leverage the generative power of LMs to generate nontoxic corpus, which achieves better data efficiency for detoxification. *2*) *Model Size*: We systematically study and mitigate the toxicity issues in LMs with parameter sizes ranging from 126M to 530B, a scale that has never been studied before in this domain. *3*) *Parameter-efficient Training*: We investigate two parameter-efficient paradigm: *adapter* [18] and *prefix-tuning* [19], and compare them with whole model adaptation in a systematic way. We hope our work can shed light on the challenges of detoxifying large-scale LMs, as well as motivate the development of detoxification techniques that are effective and parameter-efficient without significantly hurting the LM quality.

**Summary of Contributions:**

- We identify the trade-off between detoxification effectiveness (measured by Perspective API and human evaluation) and language model quality (measured by validation perplexity and downstream task accuracy). Existing approaches either suffer from limited detoxification effectiveness or significantly sacrifice the language model quality to detoxify generative LMs.

- We propose Self-Generation Enabled domain-Adaptive Training (SGEAT) that uses a self-generated dataset for detoxification. It mitigates the *exposure bias* [20, 21] from the discrepancy between teacher-forced domain-adaptive training and autoregressive generation at test time, and thus achieves better data efficiency. In particular, we demonstrate that it consistently outperforms the baseline approach with domain-adaptive training on pre-training data (DAPT) by a wide margin across various model sizes in terms of automatic and human evaluations, even when we use only a $\frac{1}{3}$ smaller corpus for training. By combining SGEAT with the state-of-the-art decoding-time method, we can further reduce the toxicity of large-scale generative LM.

- From the perspective of model size, we find that: *i)* Large LMs have similar toxicity levels as smaller ones given the same pre-training corpus. This implies the toxicity comes from the training dataset, instead of the model size. *ii)* Large LMs require more efforts (e.g., larger training corpus) to reduce toxicity.

- We explore two parameter-efficient training methods for detoxification, and observe that: *i)* domain-adaptive training with *adapter* achieves a better trade-off between toxicity and perplexity than whole model adaptation for large-scale LMs, and the improvement is more significant when the size of LMs increases; *ii) prefix-tuning* is less suitable for detoxification and demonstrates limited detoxification effectiveness and perplexity control.

We organize the rest of the paper as follows. We discuss related work in § 2 and present our evaluation protocols in § 3. We then systematically explore the domain-adaptive training with respect to training corpus in § 4, model sizes in § 5, and parameter efficiency in § 6. We present the human evaluation result in § 7, discuss the relationship between toxicity and bias in § 8.1, and conclude the paper in § 9. Some text samples can be found in Appendix D.

## 2   Related Work

Large-scale language models (LM) have achieved state-of-the-art performance on various downstream tasks. However, they also exhibit undesirable behaviors in terms of ethical, robustness, privacy, and nonfactual generation issues [10, 22–26]. For example, since they are pre-trained over a sizable

---

[4] https://www.perspectiveapi.com/.

Table 1: Evaluation of LM toxicity and quality across 5 different parameter sizes. Model toxicity is evaluated on REALTOXICITYPROMPTS benchmark through Perspective API. **Full** refers to the full set of prompts, **Toxic** and **Nontoxic** refer to the toxic and nontoxic subsets of prompts. $\downarrow$ / $\uparrow$ means the lower / higher the better. PPL is evaluated on a held-out validation set of the pre-training corpus. Utility is estimated by averaging the LM's accuracy on 9 different tasks in the zero-shot learning setting, including Lambada, BoolQ, RACE, PiQA, HellaSwag, WinoGrande, ANLI-R2, HANS and WiC. The accuracy for each task can be found in Table 9.

| Models | Exp. Max. Toxicity ($\downarrow$) | | | Toxicity Prob. ($\downarrow$) | | | Valid. | Utility |
| | Full | Toxic | Nontoxic | Full | Toxic | Nontoxic | PPL ($\downarrow$) | Avg. Acc. ($\uparrow$) |
|---|---|---|---|---|---|---|---|---|
| 126M | 0.56 | 0.76 | 0.50 | 57% | 88% | 48% | 17.76 | 46.7 |
| 357M | 0.57 | 0.78 | 0.51 | 58% | 90% | 49% | 13.18 | 50.0 |
| 1.3B | 0.57 | 0.78 | 0.52 | 59% | 90% | 51% | 10.18 | 54.3 |
| 8.3B | 0.57 | 0.77 | 0.51 | 59% | 89% | 50% | 7.86 | 60.0 |
| 530B | 0.57 | 0.77 | 0.52 | 59% | 88% | 51% | 6.27 | 64.6 |

collection of online data, they are unavoidably exposed to certain toxic content from the Internet. Recent studies [e.g., 27–29] show that pre-trained masked LMs display different levels toxicity and social biases. Another line of work focuses on the toxicity of autoregressive LMs. For instance, Wallace et al. [8] first demonstrate that synthetic text prompts can cause racist continuations with GPT-2. Gehman et al. [10] extend the analysis of LM toxicity to non-synthetic prompts, and create a benchmark dataset REALTOXICITYPROMPTS to provide a standard evaluation protocol via Perspective API to measure LM's toxicity, which is adopted by many previous work. In this paper, we follow the standard setting to compare different detoxification approaches on different-sized LMs.

**Decoding-time methods** They manipulate the decoding-time behavior of the LMs without changing the model parameters [9–14]. Simple approaches such as word filtering and vocabulary shifting [10] directly lower the probability of toxic words (e.g., swearwords, slurs, vulgar slang) being generated. Though efficient, such approaches fail to consider the semantic meaning of the generated text at the sequence level. Thus, it cannot completely prevent from generating toxic sentences which contain no undesirable words from the blocklist [15] (*e.g.,* "*poor people don't deserve to live in nice houses*"). Xu et al. [13] perform sentence-level filtering by generating $K$ continuations given the same prompt and returning the most nontoxic sentence. Similarly, Self-Debiasing [11] uses $K$ manually crafted templates to manipulate the decoding probability distribution and dynamically set the probability of toxic words to be low. However, these methods lead to $K$ times longer than the normal decoding. PPLM [9] iteratively adds perturbation on the context vector at each step of decoding. Though with better detoxification effectiveness, it suffers much more computational overhead due to multiple iterations of forwarding and backward propagation to generate the perturbations. GeDi [12] guides generation at each step with a second LM trained on nontoxic data by computing classification probabilities for all possible next tokens. However, it requires an external LM trained on non-toxic data, which is not easy to access in practice. DEXPERT [14] controls the generation of large-scale pre-trained LM with an "expert" LM trained on non-toxic data and "anti-expert" LM trained on toxic data in a product of experts [30]. It achieves the state-of-the-art detoxification results on REALTOXICITYPROMPTS, but sacrifices the validation perplexity and downstream task accuracy.

**Domain-adaptive training methods** They fine-tune the pre-trained LMs to the non-toxic domain by training on curated nontoxic data [10, 16, 31]. Gehman et al. [10] use the DAPT framework [31] to further train LMs on the nontoxic subset (filtered via the Perspective API) of pre-training corpus, OWTC, with GPT-2. Besides DAPT, Gehman et al. [10] propose to fine-tune on a corpus with toxicity attribute token and prepend the nontoxic attribute token as prompt to yield nontoxic generation. Solaiman and Dennison [16] propose a human-crafted Values-Targeted Datasets to change model behavior and reflect a set of targeted values. Baheti et al. [32] focus on mitigating the offensive behavior in dialogue systems. They leverage crowd-sourcing to label a conversation dataset generated by an existing dialogue model, and use it for offensive detection and mitigating the offensive behavior via the controlled text generation. In this work, we focus on exploring the limits of domain-adaptive training methods to reduce the toxicity of language models, while maintaining good validation perplexity and downstream task accuracy.

**Reinforcement learning (RL) methods** There are two concurrent work [33, 34] that study the toxcity behavior of LM with RL. InstructGPT [33] requires collecting human demonstrations and rankings of model outputs for two-stage fine-tunings. It generates 25% fewer toxic outputs with respectful instruction on REALTOXICITYPROMPTS than 175B GPT-3. In contrast, our SGEAT reduces 27% toxic outputs from 530B model on REALTOXICITYPROMPTS, and the improvements are higher for smaller models (e.g., reduces 37% toxic outputs from 8B model). To identify the toxic LM behavior, Perez et al. [34] uses RL to improve the generation of adversarial test cases.

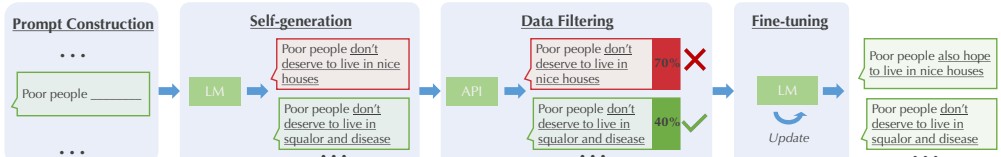

Figure 1: Overview of the SGEAT method. SGEAT constructs prompts to leverage the LMs to generate a corpus for domain-adaptive training. Then, the generated corpus is further filtered via Perspective API to ensure that the curated dataset has low toxicity. Finally, we use the filtered texts to further perform domain-adaptive training for detoxification.

## 3 Evaluation Protocols

In this section, we present our principle for evaluating different detoxification methods. Specifically, we emphasize that detoxification method should focus on both reducing the model toxicity and maintaining the model quality after detoxification. We first discuss the protocol for LM toxicity evaluation, and then present the protocol to evaluate the LM quality before and after detoxification.

**Pre-trained LMs.** We investigate the toxicity of a variety of standard GPT-3 like LMs with different parameter sizes, ranging from 126M (similar to GPT-3 Small), 357M (similar to GPT-3 Medium), 1.3B (similar to GPT-3 XL), 8.3B to the largest 530B [6]. All of the models are based on Transformer [35] with different hidden dimension, number of layers, and attention heads. We present more details in Appendix §A.1. All standard models are pre-trained on the same pre-training corpus, which is an English text corpus constructed from 15 high-quality datasets.

### 3.1 Toxicity Evaluation

In this work, we follow prior work [15, 10] and perform both automatic evaluation and human evaluation to measure an LM's tendency to generate toxic language.

**Automatic Evaluation** relies on Perspective API, an online automated model for toxic language and hate speech detection. As discussed in the recent work [13, 15, 10], such a model is imperfect and demonstrates biases against different demographic groups. Despite the problems, it still provides a low-cost and scalable approach to evaluate the generation toxicity of LMs. Moreover, both our study in Section 7 and Welbl et al. [15] find that the toxicity scores from Perspective API are strongly correlated with human evaluation, thus it is meaningful to approximately measure LM toxicity. We note that Perspective API update the models regularly. The scores returned by Perspective API may change over time. The toxicity scores reported in the following sections were evaluated before May 2022.

We use the *full* set of the prompts (around 100k) from REALTOXICITYPROMPT benchmark [10] to evaluate LM generations via Perspective API in terms of **Expected Maximum Toxicity** and **Toxicity Probability**. Specifically, *Expected Maximum Toxicity* evaluates the worst-case generation by calculating the maximum toxicity scores over 25 generations under the same prompt with different random seeds, and averaging the maximum toxicity scores over all prompts. *Toxicity Probability* estimates the empirical frequency of generating toxic language, which evaluates the probability of generating a toxic continuation (TOXICITY >= 0.5) at least *once* over 25 generations for all prompts. We follow Gehman et al. [10] and restrict the generations up to 20 tokens or below. We present the automatic evaluation of five LMs with different parameter sizes in Table 1.

**Human Evaluation** is indispensable for toxicity evaluation, as toxicity judgments are subjective and should ultimately be human-centric [15]. Specifically, we adapt the instructions from Welbl et al. [15] and ask human annotators to evaluate the continuations. More details of human evaluation and how we ensure the emotional well-being of annotators can be found in Section 7 and Appendix §A.3.

### 3.2 LM Quality Evaluation

To understand the impact of detoxification, we evaluate the quality of LM along two fronts: *perplexity* and *utility*. *Perplexity* (PPL) is evaluated on a held-out validation set of pre-training corpus [5],

---

[5]We also evaluate PPL on the filtered nontoxic portions of the validation set in Appendix §C.2. We observe the same trends of PPL increase as the full held-out validation set.

Table 2: Evaluation of LM toxicity and quality across different detoxification methods on the 1.3B LM. In the first row, ↓ / ↑ means the lower / higher the better. PPL of word banning goes to infinity as the probabilities of some banned words are set to zero. ↑ and ↓ are compared against the standard 1.3B LM. For example, ↓ is preferred for Toxicity and PPL, while ↑ is preferred for Utility Average Accuracy.

| | Models | Exp. Max. Toxicity (↓) | | | Toxicity Prob. (↓) | | | Valid. PPL (↓) | Utility Avg. Acc. (↑) |
|---|---|---|---|---|---|---|---|---|---|
| | | Full | Toxic | Nontoxic | Full | Toxic | Nontoxic | | |
| **Domain-Adaptive Training** | Jigsaw (nontoxic) | 0.58 ↑0.01 | 0.77 | 0.53 | 61% ↑2% | 90% | 53% | 11.51 ↑1.33 | 54.6 ↑0.3 |
| | DAPT (nontoxic) | 0.47 ↓0.10 | 0.69 | 0.41 | 43% ↓16% | 79% | 33% | 10.40 ↑0.22 | 54.7 ↑0.4 |
| | SGEAT (heuristic) | 0.47 ↓0.10 | 0.73 | 0.40 | 43% ↓16% | 85% | 31% | 11.14 ↑0.96 | 54.7 ↑0.4 |
| | SGEAT (standard) | 0.44 ↓0.13 | 0.67 | 0.38 | 38% ↓21% | 75% | 28% | 11.22 ↑1.04 | 54.6 ↑0.3 |
| | SGEAT (augmented) | **0.43** ↓0.14 | 0.68 | 0.37 | **37%** ↓22% | 77% | 26% | 11.19 ↑1.01 | 54.4 ↑0.1 |
| **Decoding-Time** | Word Banning | 0.54 ↓0.03 | 0.72 | 0.49 | 56% ↓3% | 86% | 47% | ∞ | 54.3 ↓0.0 |
| | Rejection Sampling (4× slow) | 0.45 ↓0.12 | 0.68 | 0.38 | 39% ↓20% | 78% | 28% | 10.18 ↑0.00 | 54.3 ↓0.00 |
| | DEXPERTS (3× slow) | 0.31 ↓0.26 | 0.50 | 0.26 | 18% ↓41% | 47% | 11% | 19.87 ↑9.46 | 46.2 ↓8.1 |
| **Combined** | SGEAT + Rejection Sampling | 0.33 ↓0.24 | 0.56 | 0.26 | 21% ↓38% | 58% | 11% | 11.19 ↑1.01 | 54.4 ↑0.1 |
| | SGEAT + DEXPERTS | **0.27** ↓0.30 | 0.45 | 0.22 | **14%** ↓45% | 40% | 7% | 20.21 ↑10.03 | 44.9 ↓9.4 |

which measures both the *fluency* and *coverage* of output language. The *utility* is estimated by the performance on downstream tasks. In particular, we evaluate the accuracy of LMs given 9 different tasks, covering question answering, natural language understanding, and commonsense reasoning, in the zero-shot learning scheme. We base the downstream tasks evaluation on Gao et al. [36]. We present the LM quality evaluation of 5 pre-trained LMs in Table 1. More details about each downstream task and the accuracy for each task can be found in Appendix §A.3.

We note some recent work [13, 15] demonstrates that existing detoxification techniques can amplify the social biases against minority groups. In this work, we mainly focus on the intrinsic quality of LM and analyze how it degrades after detoxification. We leave the bias discussion in §8.1.

In the following sections, we use above evaluation protocols to explore the limits of domain-adaptive training for detoxification on three dimensions: training corpus, model sizes, and parameter efficiency.

# 4 Impact of Training Corpus

Training corpus is a core factor that impacts the effectiveness and efficiency of domain-adaptive training. The state-of-the-art approach, DAPT [10], adopts a pre-training corpus [17] curated by Perspective API to construct the training dataset for detoxification. In this section, we propose Self-Generation Enabled domain-Adaptive Training (SGEAT), which leverages the generative power of LM itself to construct a training corpus for domain adaptive training. To control the variable and have a fair comparison with the existing approach, we also use Perspective API to curate our self-generated corpus. We show that SGEAT can further push the limits of domain-adaptive training for detoxification with better data efficiency.

## 4.1 SGEAT

As shown in Figure 1, SGEAT consists of four steps: 1) prompt construction; 2) self-generation; 3) data filtering; and 4) domain-adaptive training.

**Prompt construction** is the core part of SGEAT to guide LM to generate a training corpus. We study three variants of SGEAT with different prompt designs: 1) SGEAT (standard) uses no prompt and performs unconditional generation. 2) SGEAT (heuristic) uses a set of manually crafted prompts inspired by the definition of *toxicity* from Perspective API. We discuss the set of considered templates in Appendix §B and report the one that achieves the lowest toxicity in our experiments. 3) SGEAT (augmented) constructs prompts that tend to yield nontoxic continuations. Specifically, we find the most nontoxic documents from the unconditional generation, and split each document into half as the prompts and the continuations. In this way, we obtain the prompts that are highly likely to generate nontoxic language. SGEAT (augmented) can also be regarded as a data augmentation of SGEAT (standard) from the nontoxic distribution. We present more details in Appendix §B.

**Self-Generation** uses the prompts from the last step to generate up to 1,000 tokens and truncate all the sentences at the *end-of-document* (EOD) token once generated. We use nucleus sampling [37] with $p = 0.9$ and the temperature of 1 during generation. To demonstrate the data efficiency of SGEAT, we generate only 100k documents in total, in comparison with DAPT in Gehman et al. [10] that uses 7500k documents from the pre-training corpus.

**Data Filtering** further filters out toxic samples to ensure the training corpus is mostly nontoxic. Specifically, we follow the standard DAPT setup in Gehman et al. [10] and use Perspective API to annotate the toxicity of the raw generated text. Different from DAPT that performs aggressive filtering on pre-training data and only keeps the most nontoxic 2% of the documents, we keep the most nontoxic 50% of the generated text to demonstrate the quality and data efficiency of SGEAT. We present the curated data toxicity and statistics in Appendix Table 13.

**Domain-Adaptive Training** leverages the curated nontoxic corpus to further fine-tune the pre-trained LM with standard log-likelihood loss and adapt it to the nontoxic data domain. We present more training details in Appendix §A.2.

## 4.2 Evaluation Results of Domain-Adaptive Training

In this subsection, we evaluate existing domain-adaptive training methods on 1.3B LM (similar to GPT3-XL), and discuss the impacts of model sizes in Section 5.

**Baselines:** We consider the following domain-adaptive training baselines: **DAPT (nontoxic)** [31] uses a nontoxic subset of pre-training corpus annotated by Perspective API to perform domain-adaptive training; and **Jigsaw (nontoxic)** uses a human-annotated nontoxic subset of Jigsaw Toxic Comment Classification dataset[6].

We present the evaluation results in Table 2. Among all domain-adaptive training methods, we find that SGEAT (augmented) achieves the lowest toxicity scores with moderate perplexity increases and without degrading the LM utility accuracy (or even improving). Specifically, SGEAT (augmented) reduces the toxicity of the standard 1.3B by 0.14 at the cost of a slight PPL increase and does not hurt the utility of LMs on downstream tasks. Moreover, we note that although DAPT (nontoxic) uses 3 times larger corpus than SGEAT (augmented) (shown in Appendix Table 13), SGEAT (augmented) still achieves lower toxicity than DAPT (nontoxic), which implies that self-generated data has better data efficiency for domain-adaptive training. We think such high data efficiency comes from the fact that *i*) the self-generated corpus well captures the high-density regions of the output space of a pre-trained LM, and *ii*) training on autoregressively generated corpus mitigates the exposure bias [20, 21], which refers to the train-test discrepancy of an autoregressive model. Thus, when we train the LM on the self-generated non-toxic corpus, it tends to increase the likelihood on the non-toxic density region, which enables data-efficient training to detoxify the model.

The human-annotated nontoxic Jigsaw dataset fails to detoxify the LM and even increases the model toxicity. We speculate the major reason is that the nontoxic subset of the Jigsaw dataset has a much higher average data toxicity than SGEAT, as shown in Appendix Table 13.

Among SGEAT methods, we observe that SGEAT (augmented) achieves the best detoxification result at a similar level of PPL increase, while SGEAT (heuristic) is less effective to detoxify the LM. We think the reason lies in the data diversity: The unconditional generation covers the diverse regions of the generation distribution and yields the most diverse data distribution, and thus SGEAT (standard) also achieves good detoxification performance. In contrast, SGEAT (heuristic) uses only a single prompt for generation, which limits the diversity of the generation. More analysis about prompt design is in Appendix §B.6.

## 4.3 Evaluation Results of Decoding-time Methods

Besides the domain-adaptive training baselines, we also compare with decoding-time algorithms: **Word Banning** [10] sets the probability of generating any word from a list[7] of profanity, slurs, and swearwords to zero during decoding. **Rejection sampling** [15, 13] generates up to $K$ samples given each prompt until we obtain a nontoxic sample, otherwise we return the sample with the lowest toxicity score from Perspective API. We set $K = 4$ due to the computational limit. **DEXPERTS** [14] is the state-of-the-art decoding-time algorithm for detoxification that uses two auxiliary expert and anti-expert LMs to steer a model's generation. The expert model is the same as DAPT (nontoxic); while the anti-expert model is fine-tuned on the top toxic portion of OWTC with 150k documents.

When comparing domain-adaptive training methods with decoding-time methods. We note that rejection sampling adds $4\times$ computational overhead during decoding, but is less effective than domain-adaptive training SGEAT, as LM rarely generates nontoxic continuations given toxic prompts

---

[6] https://www.kaggle.com/c/jigsaw-toxic-comment-classification-challenge/
[7] https://github.com/LDNOOBW/List-of-Dirty-Naughty-Obscene-and-Otherwise-Bad-Words

Table 3: Evaluation of LM toxicity and quality of domain-adaptive training methods along 5 different parameter sizes. 530B$^\dagger$ is trained with more self-generated data (100k samples). 530B$^\ddagger$ is trained with more epochs (5 epochs), while the others are trained with 3 epochs. ↑ and ↓ are compared against the standard LM of the corresponding size.

| Models | | Exp. Max. Toxicity (↓) | | | Toxicity Prob. (↓) | | | Valid. PPL (↓) | Utility Avg. Acc. (↑) |
|---|---|---|---|---|---|---|---|---|---|
| | | Full | Toxic | Nontoxic | Full | Toxic | Nontoxic | | |
| **DAPT (nontoxic)** | 126M | 0.44 ↓0.12 | 0.65 | 0.38 | 37% ↓20% | 72% | 28% | 17.97 ↑0.21 | 46.0 ↓0.7 |
| | 357M | 0.47 ↓0.10 | 0.69 | 0.41 | 43% ↓15% | 78% | 33% | 13.33 ↑0.15 | 49.9 ↓0.1 |
| | 1.3B | 0.47 ↓0.10 | 0.69 | 0.41 | 43% ↓16% | 79% | 33% | 10.40 ↑0.22 | 54.7 ↑0.4 |
| | 8.3B | 0.48 ↓0.09 | 0.69 | 0.42 | 45% ↓14% | 79% | 35% | 8.12 ↑0.26 | 59.1 ↓0.9 |
| | 530B | 0.50 ↓0.07 | 0.71 | 0.45 | 49% ↓10% | 82% | 39% | 7.32 ↑1.05 | 63.4 ↓1.2 |
| **SGEAT (augmented)** | 126M | 0.39 ↓0.17 | 0.63 | 0.33 | 30% ↓27% | 69% | 19% | 19.55 ↑1.79 | 46.3 ↓0.4 |
| | 357M | 0.42 ↓0.15 | 0.68 | 0.35 | 36% ↓22% | 77% | 24% | 14.39 ↑1.21 | 49.3 ↓0.7 |
| | 1.3B | 0.43 ↓0.14 | 0.68 | 0.37 | 37% ↓22% | 77% | 26% | 11.19 ↑1.01 | 54.4 ↑0.1 |
| | 8.3B | 0.44 ↓0.13 | 0.68 | 0.37 | 38% ↓21% | 76% | 28% | 8.91 ↑1.05 | 59.1 ↓0.9 |
| | 530B | 0.46 ↓0.11 | 0.70 | 0.40 | 43% ↓16% | 80% | 32% | 7.86 ↑1.59 | 62.6 ↓2.0 |
| | 530B$^\dagger$ | 0.45 ↓0.12 | 0.69 | 0.39 | 41% ↓18% | 78% | 31% | 7.92 ↑1.65 | 62.0 ↓2.6 |
| | 530B$^\ddagger$ | 0.44 ↓0.13 | 0.67 | 0.38 | 39% ↓20% | 76% | 29% | 9.63 ↑3.36 | 58.8 ↓5.8 |

[13]. Although the state-of-the-art DEXPERTS achieves significantly lower toxicity scores than SGEAT, we also observe that there is a concerning perplexity and utility degradation, with an increase of 9.47 in PPL and a drop of 9.4% in downstream task accuracy. Such degradation makes the detoxified 1.3B LM quality even worse than a standard 126M LM, as shown in Table 1. We hope that our findings can motivate researchers to focus more on the trade-off between detoxification and LM quality when designing detoxification algorithms. Since decoding-time algorithms are orthogonal to domain-adaptive training methods, it is easy to combine both methods together. Specifically, we replace the standard 1.3B model used in rejection sampling and DEXPERTS with SGEAT (augmented) detoxified one, and observe that the combined method can yield the lowest toxicity scores among existing methods.

# 5 Impact of Model Size

We next investigate how the number of model parameters impacts the domain-adaptive training for detoxification. Specifically, we show that 1) models with different number of parameters trained on the same pre-training corpus display similar levels of toxicity; 2) self-generated data consistently demonstrates better detoxification effectiveness than pre-training corpus across different parameter sizes; 3) larger LMs require more efforts to reduce the toxicity.

**Standard Model Toxicity.** We first evaluate the toxicity of 5 standard LMs across different parameter sizes in Table 1 and Table 9. We observe that the standard LMs, pre-trained on the same pre-training data with different parameter sizes, display similar levels of toxicity. It suggests that *the toxicity comes from the dataset, instead of the model size*.

**Detoxification Effectiveness of SGEAT.** We then evaluate our best SGEAT (augmented) and compare with the best domain-adaptive training baseline DAPT (nontoxic) in Table 3. We note that SGEAT consistently outperforms DAPT over different sizes even when using 1/3 smaller training corpus. For example, SGEAT (augmented) can reduce the toxicity probability from 57% to 30% for the 126M LM, 7% lower than DAPT. These results confirm that: *the self-generated corpus is more efficient to detoxify the LM than using the curated corpus of pre-training data.*

**Larger-scale LMs requires more endeavors to detoxify.** From Table 3, we observe the detoxification effectiveness decays for both DAPT and SGEAT with the increase of LM parameter sizes. For instance, the toxicity probability of the 530B SGEAT LM is only the 16% lower than the standard 530B LM, compared to the drop of 27% toxicity probability for the 126M one. We figure the potential reason of such small improvement on larger LM is that large LM tends to require more training data and fine-tuning epochs to detoxify. Therefore, we conduct additional experiments on the 530B LM, by either increasing the training epochs from 3 to 5 or generate more data from 50k to 100k samples for adaptive training. We find that while both methods further reduce the toxicity of the 530B LM, training for more epochs might lead to model overfitting and hurts the PPL and downstream accuracy by a large margin. In contrast, training with more data demonstrates a better trade-off between detoxification and LM quality. It implies that *it needs more endeavors to detoxify large-scale LMs.*

Table 4: Evaluation of LM toxicity and perplexity of parameter-efficient training methods. ↑ and ↓ are compared against whole model adaptation. We conduct this ablation study using DAPT (nontoxic).

(a) Adapter [18]

| Projection Size | Toxicity (↓) | | Valid. PPL (↓) |
|---|---|---|---|
| | Exp. Max. Toxicity | Toxicity Prob. | |
| 256 | 0.49 ↑0.02 | 46% ↑3% | 10.34 ↓0.06 |
| 512 | 0.49 ↑0.02 | 45% ↑2% | 10.36 ↓0.04 |
| 1024 | 0.48 ↑0.01 | 45% ↑2% | 10.39 ↓0.01 |

(b) Prefix Tuning [19]

| Prefix Length | Toxicity (↓) | | Valid. PPL (↓) |
|---|---|---|---|
| | Exp. Max. Toxicity | Toxicity Prob. | |
| 128 | 0.51 ↑0.04 | 49% ↑6% | 10.35 ↓0.05 |
| 256 | 0.51 ↑0.04 | 48% ↑5% | 10.45 ↑0.05 |
| 512 | 0.52 ↑0.05 | 50% ↑7% | 10.56 ↑0.16 |

Table 5: Evaluation of LM toxicity and quality of adapter for large-scale LMs. ↑ and ↓ are compared against whole model adaptation.

| Models (Projection Size=1024) | | Exp. Max. Toxicity (↓) | | | Toxicity Prob. (↓) | | | Valid. PPL (↓) | Utility Avg. Acc. (↑) |
|---|---|---|---|---|---|---|---|---|---|
| | | Full | Toxic | Nontoxic | Full | Toxic | Nontoxic | | |
| DAPT (nontoxic) | 8.3B | 0.48 ↓0.00 | 0.70 | 0.42 | 45% ↓0% | 79% | 36% | 7.99 ↓0.13 | 59.4 ↑0.3 |
| +adapter | 530B | 0.50 ↓0.00 | 0.71 | 0.45 | 49% ↓0% | 82% | 40% | 6.69 ↓0.63 | 63.7 ↑0.3 |
| SGEAT (augmented) | 8.3B | 0.44 ↓0.00 | 0.68 | 0.37 | 38% ↓0% | 77% | 28% | 8.88 ↓0.03 | 59.0 ↓0.1 |
| +adapter | 530B | 0.46 ↓0.00 | 0.69 | 0.39 | 41% ↓2% | 79% | 31% | 7.22 ↓0.64 | 63.3 ↑0.7 |

**LM Quality Evaluation.** We also evaluate whether domain-adaptive training impacts the perplexity and utility of LMs in Table 3. When trained within 3 epochs, we find that the PPL of LMs slightly increases and the LM utility drops a little in most cases, which suggest that *models gradually adapt to the nontoxic domain without a significant sign of overfitting or degradation in terms of LM quality.*

**Domain Adaptation v.s. Overfitting.** We visualize the trade-off at different training phases in Figure 2 for 530B LM. Specifically, we record the validation perplexity and model toxicity after 1, 3, and 5 training epochs for *DAPT (nontoxic, 150k)* and *SGEAT (augmented, 50k)*. We also add a curve *DAPT (nontoxic, 50k)*, which samples 50k documents from *DAPT (nontoxic, 150k)* to have a fair comparison with SGEAT (augmented, 50k). We observe that at the beginning of training, the model toxicity drops substantially and barely sacrifices the model PPL (steep slope). Then it is gradually adapted towards the nontoxic domain. SGEAT demonstrates a better trade-off between toxicity and quality, as SGEAT achieves substantially lower toxicity with the same PPL after 1 epoch of training. Finally, we observe the curve

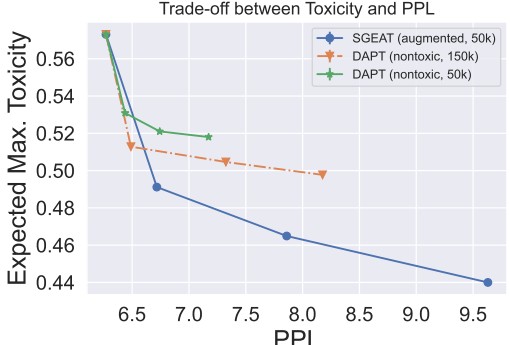

Figure 2: The expected maximum toxicity v.s. model perplexity for the 530B LM at different training steps.

is becoming more flat, especially for DAPT, which indicates the transition from the domain adaptation to overfitting.

For LMs with different sizes fine-tuned with different methods, we find 3 epochs is a good cut-off point for whole model adaption, which achieves good trade-off between model toxicity and perplexity. This rule of thumb is also aligned with previous study [10].

## 6 Parameter-efficient Training

To cope with the challenges of large-scale LMs, we explore two parameter-efficient training paradigms: *adapter* [18] and *prefix tuning* [19], and evaluate whether they can improve the LM quality and achieve a better trade-off between detoxification and LM quality than whole model adaption. We show that: in the scenario of detoxification, 1) adapter demonstrates a better trade-off than prefix tuning, and 2) adapter can further mitigate the drop of LM quality and improve the trade-off upon whole-model adaptation for large-scale LMs.

### 6.1 Comparison between Adapter and Prefix Tuning

Both adapter and prefix tuning add additional parameters to the standard LM, and only optimize the added parameters during training without perturbing the original LM parameters. Such paradigm provides the flexibility, especially for large-scale LMs, to adapt to different domains with a few additional parameters, rather than heavily fine-tune the whole model with multiple copies of the

whole model parameters for different domains. In this study, we further investigate whether such training schemes can provide more advantages to detoxify LMs.

*Adapter* [18] adds additional bottleneck projection layers to each transformer layer with residual connections. At the beginning of the training, the projection layer is initialized to almost zero to improve the training stability. *Prefix tuning* [19] appends additional continuous "prefix" vectors to the input to better steer LMs' generations. To have a comprehensive understanding and comparison between adapter and prefix tuning, we first perform ablation studies on small-scale 1.3B LM over the key hyper-parameters: the projection size for adapter and the prefix length for prefix tuning. We follow the same training schedules as whole model adaptation but train more epochs so that the PPL reaches a similar level as whole model adaptation. We present the evaluation results in Table 4.

When comparing Table 4a with Table 4b, we observe that adapter demonstrates a better trade-off between detoxification and LM quality than prefix tuning. We figure the possible reasons are two folds: 1) given the same projection size and prefix length, the number of additional parameters of adapter is around twice more than prefix tuning, which gives more capacity for adapter to perform domain adaptation; 2) however, while longer prefix length could give more capacity to steer the model generation, it also adds too many irrelevant contexts, which not only hurts the perplexity of the LM but also slows down the decoding speed. Compared to the whole model adaption, adapter does not show significant advantages in terms of detoxification and LM quality for small-scale models like 1.3B one. For adapter results with different projection sizes, we observe that a larger projection size yields better detoxification effectiveness possibly due to larger model capacity. We thus apply adapter with the projection size=1024 to larger-scale LMs (8.3B and 530B) and investigate whether it can solve the challenges of large-scale LMs.

## 6.2 Apply Adapter to larger-scale Models

We follow the same training schedules as the whole model adaptation to train the adapters for larger-scale LMs. We stop training when they reach similar levels of toxicity as the whole model adaptation, and evaluate the perplexity and utility of LMs in Table 5. We can see that for larger-scale LMs, adapter can not only improve the parameter efficiency, but also mitigate the PPL and the LM quality drop. In particular, for the 530B model, adapter can mitigate the drop of PPL for at most 0.64 and improve the average downstream task accuracy by 0.7%.

## 7 Human Evaluation

We further verify our findings via human evaluation on the standard models, DAPT, SGEAT, and decoding-time algorithm DEXPERTS across five LM sizes.

**Setup.** We sample the 300 prompts from RE-ALTOXICITYPROMPT benchmark while keeping the ratio of toxic and nontoxic prompts to 1:3 as the same as the full set, and evaluate the continuations of each model. We follow Welbl et al. [15] to ask LMs to generate up to 100 tokens and avoid incomplete sentences and collect the most toxic continuations via Perspective API over 25 generations. Finally, we gather 5,700 continuations from 19 models and randomly shuffle them for human evaluation. Then we group samples into a batch of 10, and assign them to 5 annotators. In total 187 workers from Amazon MTurk participated in the evaluation. To consider the annotators' well-being, we make sure the average number

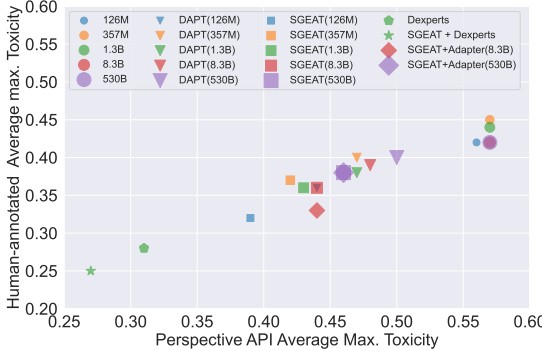

Figure 3: (best viewed in color) Average human toxicity scores v.s. Perspective API scores for the different methods we evaluate. The Pearson correlation coefficient is 0.9661.

of toxic samples (TOXICITY >= 0.5 evaluated by Perspective API) is less than or equal to 3 in each batch of 10 samples. To calculate the average scores of annotations, we follow Welbl et al. [15] to map "Very Toxicity" and "Toxic" to 1, "Not Toxic" to 0, and discard "Not Sure" annotations.

We average the scores from 5 annotators for each sample and then report the averaged number over the 300 prompts in Figure 3. The detailed scores can be found in Table 8 in Appendix. We present more details in Appendix §A.3. By comparing the objective evaluation with human evaluation,

Table 6: LM PPL in the gender and ethinicity domains on the BOLD dataset. ↑: based on standard 1.3B LM.

| Models | Gender (↓) | | Ethnicity (↓) | | | |
|---|---|---|---|---|---|---|
| | Male | Female | European | Asian | African | Hispanic |
| Standard | 11.6 | 11.4 | 13.9 | 13.5 | 14.1 | 15.6 |
| SGEAT | 12.7 ↑1.1 | 12.4 ↑1.0 | 15.1 ↑1.2 | 14.8 ↑1.3 | 15.4 ↑1.3 | 17.2 ↑1.6 |

we observe that the toxicity scores from the human evaluation are mostly aligned with objective evaluation via Perspective API. Such findings are also confirmed by Welbl et al. [15]. The human evaluation also verifies that $i$) LMs of different sizes have similar levels of toxicity, and $ii$) LMs of larger sizes present more challenges to detoxify.

## 8 Discussion

### 8.1 Bias against Marginalized Groups

We follow the setting of Welbl et al. [15] and evaluate the PPL of the 1.3B standard LM and SGEAT (augmented) fine-tuned LM on the *gender* and *ethnicity* domains using the BOLD dataset [38] as shown in Table 6. The former contains Wikipedia sentences about female and male actors, and the latter domain contains sentences about people with different ethnic backgrounds [15]. We find that: (*i*) LM PPL increases moderately on the BOLD dataset after effective detoxification, which is aligned with our findings in §4.2. (*ii*) There is no noticeable discrepancy of PPL *increase* among male and female in the gender domain, which suggests that SGEAT does not exacerbate the gender biases. (*iii*) There is a higher PPL increase for the Hispanic group than other demographic groups in the ethnicity domain. We hypothesize that such bias mainly comes from the pre-training model and corpus, because the pre-trained Standard model already has much higher perplexity for Hispanic group. Our findings partly align with recent findings on the trade-off between detoxification and bias [13, 15]. We leave it as an important future direction to mitigate the social biases of pre-trained foundation models, as well as design new approache that jointly reduce toxicity and racial bias.

### 8.2 Limitation of SGEAT

While we observe that SGEAT has demonstrated very good trade-off between detoxification effectiveness and perplexity, SGEAT still has potentials to further improve.

**Bias within Hate Speech Detector.** Similar to DAPT, SGEAT also relies on a hate speech classifier (*i.e.,* Perspective API) to filter out toxic samples. However, existing classifier on toxicity classification is imperfect and is known to amplify the social bias against different demographic groups due to the annotation bias and sampling bias [13] (*e.g.,* the classifier tend to assign higher toxicity scores for text mentioning historically underrepresented groups). As a result, SGEAT may also be impacted due to the use of Perspective API, which may filter both toxic text and minority identity mentions. Nevertheless, we believe that SGEAT can get more benefits with a more robust, unbiased, and fair hate speech detector, so models fine-tuned on the filtered corpus can unlearn toxicity without forgetting corpus from minority groups.

**Bias within Pre-trained Model.** As discussed in § 8.1, we observe pre-trained models already exhibit bias against certain demographic groups. As a result, the self-generated corpus may inherit the bias and harm the coverage of detoxification. Thus we leave it as an important future direction to build a bias-free pre-trained LM, which can benefit SGEAT and other detoxification methods.

## 9 Conclusion

We explore the limits of domain-adaptive training for detoxifying LMs along three aspects: 1) training corpus; 2) model size and 3) parameter-efficient training. We first identify the trade-off between detoxification effectiveness and LM quality in detoxification methods. We propose Self-Generation Enabled domain-Adaptive Training (SGEAT), which leverages the generative power of LMs for data-efficient and effective detoxification. We comprehensively detoxify LMs with parameters sizes ranging from 126M up to 530B and find interesting properties of large-scale LMs. We demonstrate that *adapter* provides parameter-efficient training and achieves a better trade-off of toxicity and LM quality. We hope our work can shed light on the development of detoxification techniques that can largely reduce toxicity while maintaining good perplexity and downstream task accuracies.

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
