# OpenReview forum: "Exploring the Limits of Domain-Adaptive Training for Detoxifying Large-Scale Language Models"
_NeurIPS.cc/2022/Conference — NeurIPS 2022 Accept_

### Official Review · Reviewer_zpdX · 2022-07-10

**Rating:** 6
**Confidence:** 4
**Soundness:** 3 good
**Presentation:** 3 good
**Contribution:** 3 good

**Summary:**

This paper investigates the domain-adaptive training for detoxifying while maintaining LM quality of language models. They focus on three points: training corpus, model size, and parameter efficiency. They've made the contributions and concluded the important observations as follows:
1) from the perspective of training corpus, the proposed self-generated dataset is more helpful for detoxification compared with domain-adaptive training on nontoxic subset of pre-training data.
2) from the perspective of model size, larger LMs suffer more from toxicity from the training corpus
3) from the perspective of parameter efficiency, adapter achieved better trade-off between toxicity and perplexity compared with prefix-tuning.

**Questions:**

Q1: the model architectures of different scales are not clearly described. From Section 3, the list of parameter sizes of different models are introduced, but are they share a similar architecture and what kind of architectures do they adopt?

Q2: In related work, domain-adaptive training methods, such as Gehman et al. (prepend nontoxic attribute token as prompt) and  Solaiman and Dennison [16] (change model behavior & reflect a set of targeted values), are discussed. What's the performance of these approaches compared with the proposed SGEAT?

Q3: Besides domain-adaptive training and decoding-time approaches, RL approaches such as InstructGPT have been proposed to avoid toxicity of LMs. What's the performance of RL-based approaches compared with the proposed SGEAT?

Q4: in Table 2, it seems combining domain-adaptive training approach (SGEAT) and decoding-time approach (Rejection Sampling and DExperts) can further improve toxicity evaluation and utility evaluation. How about the combination of other domian-adaptive training approaches, e.g., Jigsaw and DAPT, with decoding-time approach?

Q5: as shown in Tbale 3, SGEAT obtains better performance on toxicity mitigation while DAPT achieves better results on perplexity and utility evaluation. How about the combination of two, i.e., fine-tuning the LMs on the dataset containing both nontoxic pre-training corpus and generated text data? Can the combined corpus take the advantage of both strategies and therefore have a better trade-off?

**Limitations:**

The authors have discussed the limitations of this paper in appendix, from the perspective of:
- limitation of toxicity evaluation platform Perspective API;
- potential bias in detoxification methods is left for future work

**Strengths And Weaknesses:**

Strengths:

1) originality:
- this paper proposes the self-generated dataset for detoxification, which brings benefits to mitigating toxicity compared with baselines that fine-tuning LMs on nontoxic pre-training corpus
- this paper studies effectiveness of different detoxifying approaches on LMs of diverse scales, which include 530B model that hasn't been studied for toxicity before.

2) quality & clarity:
- the paper is well written and declares points clearly in each section
- the performance of different strategies and their improvement/degradation are clearly reported; results of experiments with multiple random seeds are also reported in appendix

3) significance:
- strategies compared and observations discovered in this paper may shed light on follow-up work in improving trade-off between toxicity and utility, large-scale LM design including architecture, training corpus for pertaining, etc.

Weaknesses
1) clarity
- the experiments focus on generative language models, including toxicity evaluation with RealToxityPrompt benchmark and zero-shot utility evaluation on downstream tasks following the prior framework. However, the title, abstract, introduction and conclusion haven't discussed this point, but use "language model" instead. This could be confusing and easily leads to the conclusion that the observation discovered in this paper could apply to general language models.

2) significance:
- fine-tuning LMs with untoxic generated text data is the core idea in the proposed SGEAT. However, according to the results reported in Table 2 and 3, although SGEAT could improve performance on toxicity evaluation, the performance on LM quality (perplexity and accuracy on downstream tasks) is consistently poorer than the baseline approaches DAPT and Jigsaw. Those experiments do not provide clear guidance on which kind of adaptation approach is better for projects that would like to obtain a trade-off between the two.

- Toxicity is an important problem when fine-tuning not only generative LMs, but also general (encoder alone) LMs on downstream tasks. The proposed strategy and evaluation conducted in this paper could be more significant if general LMs are also studied.

---

> ### Author Response · Authors · 2022-08-02
> **Response to Reviewer zpdX (Part 1)**
>
> Many thanks for your detailed review and comments. The suggestions are very helpful to improve the quality of our paper. We will address your comments in the following.
>
> 1, “the experiments focus on generative language models…However, the title, abstract, introduction and conclusion haven't discussed this point, but use "language model" instead.”
> * Thanks for your suggestion. We have revised our paper in the abstract and introduction and use the term of the generative language models to make it more clear.
>
> 2, “However, according to the results reported in Table 2 and 3, although SGEAT could improve performance on toxicity evaluation, the performance on LM quality (perplexity and accuracy on downstream tasks) is consistently poorer than the baseline approaches DAPT and Jigsaw. Those experiments do not provide clear guidance on which kind of adaptation approach is better for projects that would like to obtain a trade-off between the two?”
> * Thank you for your valuable comment. We want to mention that although the perplexity of SGEAT is worse than DAPT, the downstream task accuracy is mostly similar to or even better than DAPT.  Moreover, applying adapters is helpful to mitigate the drop of LM quality.
> * In terms of the guidance on which adaptation methods to choose, we summarize here: 1) if people focus more on the toxicity issues and allow minor valid PPL drop, we suggest that using SGEAT with adapters can be an efficient method to achieve a good trade-off between toxicity and PPL; 2) however if valid PPL is very important in some applications, we suggest that mixing DAPT with SGEAT and performing parameter-efficient training can achieve better PPL and satisfactory toxicity as well. We add the discussion in the revision in Appendix D.1.
>
> 3, “Toxicity is an important problem when fine-tuning not only generative LMs, but also general (encoder alone) LMs on downstream tasks. The proposed strategy and evaluation conducted in this paper could be more significant if general LMs are also studied.”
> * Thank you for pointing out this interesting direction. In our work, we mainly follow the recent work [1,2,3] and focus on the natural language generation (NLG) task, which mainly uses autoregressive generative LM. It is possible that encoder-alone LMs can output toxic content when performing cloze-style generation. We leave it as an interesting future direction and look forward to seeing more efforts from the community in detoxifying general LMs and proposing more benchmarks suitable for encoder-alone LMs.
>
> [1] Gehman, Samuel et al. “RealToxicityPrompts: Evaluating Neural Toxic Degeneration in Language Models.” Findings in EMNLP (2020)
>
> [2] Welbl, Johannes et al. “Challenges in Detoxifying Language Models.” Findings in EMNLP (2021)
>
> [3] Ouyang, Long et al. “Training language models to follow instructions with human feedback.” ArXiv abs/2203.02155 (2022).
>
> **Questions:**
>
> Q1: "the model architectures of different scales are not clearly described. From Section 3, the list of parameter sizes of different models are introduced, but are they share a similar architecture and what kind of architectures do they adopt?"
> * We describe the model architectures in Appendix A.1. All the models share the same model architects (decoder-only transformer) as GPT-3 [4], but have a different number of layers, hidden sizes, and attention heads. We specify the details in Appendix Table 6.
> [4] Brown, Tom B. et al. “Language Models are Few-Shot Learners.”  (2020)
>
> Q2: “In related work, domain-adaptive training methods, such as Gehman et al. (prepend nontoxic attribute token as prompt) and Solaiman and Dennison [16] (change model behavior & reflect a set of targeted values), are discussed. What's the performance of these approaches compared with the proposed SGEAT?”
> * As shown in [1], Atcon (Gehman et al.) is less effective than DAPT in terms of detoxification, while SGEAT achieves better detoxification results than DAPT. Solaiman and Dennison [16] do not evaluate their model on the RealToxicityPrompt benchmark and do not open source their code and datasets. So we are unable to compare with their methods.
>
> Q3: “Besides domain-adaptive training and decoding-time approaches, RL approaches such as InstructGPT have been proposed to avoid toxicity of LMs. What's the performance of RL-based approaches compared with the proposed SGEAT?”
> * We discuss the RL-based approaches in Line 114-120 in Section 2. InstructGPT is shown to generate 25%  fewer toxic outputs with respectful instruction on RealToxicityPrompt than 175B GPT-3. In contrast, our SGEAT reduces 27% toxic outputs from 530B model on RealToxicityPrompt, and the improvements are higher for smaller models (e.g., reduces 37% toxic outputs from 8B model).

---

> > ### Comment · Reviewer_zpdX · 2022-08-08
> > **About Response to Q1 and comment 2**
> >
> > Thanks for the model architecture clarification in response to Q1.
> >
> > About the response to comment 2: I think the performance on ppl and accuracy is consistent, we can hardly come to the conclusion that SGEAT is competitive or even better than DAPT in terms of downstream accuracy.

---

> > > ### Author Response · Authors · 2022-08-09
> > > **Thank you so much for your feedback**
> > >
> > > We really appreciate your comments and feedback.
> > >
> > > Thanks for pointing it out. We agree that the performance on validation PPL and downstream accuracy is consistent. Many thanks for your valuable inputs again!

---

> ### Author Response · Authors · 2022-08-02
> **Response to Reviewer zpdX (Part 2)**
>
> **Questions (continued):**
>
> Q4: “in Table 2, it seems combining domain-adaptive training approach (SGEAT) and decoding-time approach (Rejection Sampling and DExperts) can further improve toxicity evaluation and utility evaluation. How about the combination of other domian-adaptive training approaches, e.g., Jigsaw and DAPT, with decoding-time approach?”
> * Thank you for your valuable suggestion. We follow your comments and evaluate the detoxification effectiveness and PPL of combining the best domain-adaptive training baseline DAPT and the best decoding-time approach Dexperts. The results are shown in the table below. We find that combining DAPT with DExperts not only achieves higher toxicity, but also yield higher perplexity than combining SGEAT with DExperts.
>
> | | SGEAT + Dexperts | DAPT + Dexperts | Dexperts |
> |------------| -----------|------------|------------|
> | Exp. Max. Toxicity | 0.27 |  0.30 | 0.31 |
> | PPL | 20.21 |   20.32 | 19.87 |
>
>
>
> Q5: “as shown in Table 3, SGEAT obtains better performance on toxicity mitigation while DAPT achieves better results on perplexity and utility evaluation. How about the combination of two, i.e., fine-tuning the LMs on the dataset containing both nontoxic pre-training corpus and generated text data? Can the combined corpus take the advantage of both strategies and therefore have a better trade-off?”
> * Thank you for your constructive suggestion. We follow your suggestion and fine-tune the mixed dataset of DAPT and SGEAT (augmented) with the mixture ratio of 0.5. We follow the same training schedules and iterations of DAPT (nontoxic) to fine-tune the LM on the mixed dataset. The results are shown below.
> | | SGEAT (augmented) | DAPT (nontoxic) | Mix of SGEAT and DAPT |
> |------------| -----------|------------|------------|
> | Exp. Max. Toxicity | 0.43 |  0.47 | 0.46 |
> | PPL | 11.19 |   10.40 | 10.67 |
>
> * From the results, we can see that the detoxification effect of the mixed SGEAT and DAPT is an interpolation of both methods. We agree that combining these two methods does yield a better trade-off, and provides a flexible way to control the PPL and toxicity by controlling different mixture ratios. We add more discussion in the updated manuscript in Appendix C.5.

---

> > ### Comment · Reviewer_zpdX · 2022-08-08
> > **About Response to Part 2**
> >
> > I think the results from the supplemented experiments make sense. Thank you.

---

### Official Review · Reviewer_Pnxe · 2022-07-10

**Rating:** 6
**Confidence:** 4
**Soundness:** 2 fair
**Presentation:** 3 good
**Contribution:** 3 good

**Summary:**

This work explores domain-adaptive training to reduce the toxicity of language models in terms of training corpus, model size, and parameter efficiency.

As the results show, for the training corpus, using the proposed Self-Generation Enabled domain-Adaptive Training (SGEAT) that uses a self-generated dataset for detoxification gives better performance compared to the existing baselines. For the model size, this work shows that when given the same pre-training corpus, the toxicity level is still the same across different model sizes. Where the authors claim that this finding implies that the toxicity comes from the training set, instead of the model size. Moreover, they also show that large language models are harder to detoxify compared to smaller ones. Finally, this work also demonstrates that adding adapter layers to standard language models for parameter-efficient training improves the model perplexity while maintaining the same level of detoxification.

**Questions:**

Please refer to the main weaknesses section.

**Limitations:**

1. The main limitation of this work is the error propagation from the automatically generated training set and the Prespective API that has been used to filter the non-toxic samples.
2. Please see the other limitation in the main weaknesses section.

**Strengths And Weaknesses:**

Strengths:
1. This work studies an important problem of toxicity evaluation and mitigation in language models.
2. The proposed SGEAT approach that uses language models to automatically generate non-toxic training data seems promising when the generated data are used to fine-tune/detoxify the same language model compared with using the time-consuming human-annotated non-toxic datasets.
3. Extensive experiments and ablation studies have been carried out to support the proposed methods.
4. The authors have not only evaluated the detoxification performance, but also the quality and utility of the detoxified language models.
5. The findings regarding the relationship between toxicity level, training data, and model size are very interesting and can benefit the community.

Weaknesses:
Major:
1. This work did not study the detoxification performance of the generated training set with SGEAT when used to fine-tune other language models different from the one it was generated on.
2. The authors addressed in detail the effect of fine-tuning/detoxifying with different epochs, but did not do the same detailed evaluation on training samples. For instance, it would be interesting to also have the same Figure 2 for different training samples.
3. In lines 299-300, the authors claim that "SGEAT achieves substantially lower toxicity with the same perplexity" while Figure 2 shows a different trend.
4. In Table 5, the combination of SGEAT and adapter has only been evaluated on the perplexity and utility of the language models but not on the detoxification/toxicity level. It would be interesting to also evaluate the effect of this combination on toxicity.
5. Since there are no prompts for the SGEAT (standard), how is it controlled so that it does not randomly generate duplicated outputs? Has this been evaluated?
6. Why is SGEAT (augmented) only generated from the non-toxic samples of SGEAT (standard), but not also from non-toxic samples of SGEAT (heuristic)?

Minor:
1. SGEAT and SGEFT are used interchangeably throughout the paper including Figure 2 and Table 12 while SGEFT was not defined anywhere in the paper. Is it a typo?

---

> ### Author Response · Authors · 2022-08-02
> **Response to Reviewer Pnxe (Part 1)**
>
> Many thanks for your detailed comments & suggestions. They are very helpful to improve the quality of our paper. We will address your comments in the following.
>
> **Major**:
>
> 1, “This work did not study the detoxification performance of the generated training set with SGEAT when used to fine-tune other language models different from the one it was generated on.”
> * Thank you for your valuable comments. We follow your suggestion and fine-tune a 126M model with the 1.3B generated corpus SGEAT (augmented) following the same training strategy. We evaluate the expected maximum toxicity and perplexity and compare with the 126M model fine-tuned with 126M generated corpus SGEAT (augmented). The results are shown below.
>
> | 126M Model | SGEAT (augmented, 126M) | SGEAT (augmented, 1.3B) |
> |------------| -----------|------------|
> | Exp. Max. Toxicity | 0.39 |  0.41 |
> | Valid PPL | 19.55 |   18.76 |
>
> In terms of toxicity reduction, we observe that using the generated corpus from a larger LM to fine-tune a smaller LM is not as effective as using the self-generated corpus, which emphasizes the importance of fine-tuning with self-generated data to mitigate the exposure bias. However, the corpus generated from 1.3B LM does have better language quality than 126M and is closer to the pre-training corpus, thus leading to a better validation PPL than the self-generated corpus. We add more discussion in the revision in Appendix C.4.
>
> 2, “The authors addressed in detail the effect of fine-tuning/detoxifying with different epochs, but did not do the same detailed evaluation on training samples. For instance, it would be interesting to also have the same Figure 2 for different training samples.”
> * Thank you for the valuable suggestion. We follow your suggestion and sample a 1/3 subset of OWTC, which consists of 50K nontoxic documents (the same number of documents as SGEAT (augmented)). We fine-tune the 530B LM with the sampled nontoxic OWTC dataset. We evaluate the toxicity, and its corresponding validation PPL as in the updated Figure 2. We also list the results in the table below.
>
> | SGEAT (augmented, 50k) | 1 epoch | 3 epoch | 5 epoch |
> |------------| -----------|------------|------------|
> |Exp. Max. Toxicity | 0.491 |  0.465 | 0.440 |
> |PPL | 6.72 |   7.86 |  9.63 |
>
> | DAPT (OWTC, 50k) | 1 epoch | 3 epoch | 5 epoch |
> |------------| -----------|------------|------------|
> |Exp. Max. Toxicity | 0.531 | 0.521 | 0.518 |
> |PPL | 6.438 | 6.74 | 7.17 |
>
> * We note that the curve of DAPT (OWTC, 50k) is similar to SGEAT (augmented, 50k), where the model toxicity first drops significantly (steep slope), then gradually adapts towards the nontoxic domain, and finally becomes flatter. We also observe that reducing the size of the dataset makes DAPT less efficient in detoxifying 530B LM, which further confirms that large-scale LMs require more endeavors (e.g., more training data) to detoxify. We add the discussion in the revision in Section 5.
>
> 3, “In lines 299-300, the authors claim that "SGEAT achieves substantially lower toxicity with the same perplexity" while Figure 2 shows a different trend.”
> * Thanks for raising the question. This original claim was referring to the trend after 1 epoch of training. In Figure 2, we observe that SGEAT indeed achieves a lower toxicity than DAPT (nontoxic) given the same valid PPL after 1 epoch of training. This trend becomes more obvious, and the performance gap is larger when we compare SGEAT with DAPT (nontoxic, 50k) in the updated Figure 2, which is a more fair comparison given the same number of training samples. We have clarified the claim in the revision in Section 5.
>
> 4, “In Table 5, the combination of SGEAT and adapter has only been evaluated on the perplexity and utility of the language models but not on the detoxification/toxicity level. It would be interesting to also evaluate the effect of this combination on toxicity.”
> * We want clarify that we do evaluate SGEAT and adapter and compare them with the whole model adaptation on the **same detoxification/toxicity level** in Table 5 (the ↓  is compared with the whole model adaptation, instead of the standard LMs before detoxification). We observe that on the same detoxification level, parameter-efficient training is particularly helpful to mitigate the perplexity increase.

---

> ### Author Response · Authors · 2022-08-02
> **Response to Reviewer Pnxe (Part 2)**
>
> **Major (continued):**
>
> 5, “Since there are no prompts for the SGEAT (standard), how is it controlled so that it does not randomly generate duplicate outputs? Has this been evaluated?”
> * Thank you for the valuable comments. During generation, we use top_p sampling to be 0.9 with different random seeds, which significantly reduces the probability to generate duplicated output. Specifically, this setting will have on average more than 200 candidate tokens to sample at each step, and we generate up to 1000 steps. Thus the likelihood of generating duplicated data should be very small. To further verify the findings, we evaluate the diversity of SGEAT (standard) and OWTC using distinct-1, distinct-2, distinct3 & distinct-4, which measures the number of distinct n-grams of the corpus [1]. The results are shown in the following Table.
> | Methods | distinct-1 | distinct-2 | distinct-3 | distinct-4 |
> |------------| -----------|------------|------------|------------|
> |SGEAT(standard) | 0.039 | 0.282 |  0.615 | 0.828  |
> | OWTC | 0.049 | 0.336 | 0.670 | 0.854 |
>
> * We find that the diversity of SGEAT (standard) is close to the real-world corpus OWTC. We add more discussion in Section B.5.
>
> [1] Li, Jiwei et al. “A Diversity-Promoting Objective Function for Neural Conversation Models.” NAACL (2016).
>
>
> 6, “Why is SGEAT (augmented) only generated from the non-toxic samples of SGEAT (standard), but not also from non-toxic samples of SGEAT (heuristic)?”
> *  As shown in Table 2, SGEAT (standard) has lower toxicity scores than SGEAT (heuristic). We also evaluate the diversity of SGEAT (standard) and SGEAT (heuristic) using distinct-1, distinct-2, distinct3 & distinct-4 as shown in the table below, and observe that SGEAT (heuristic) is less diverse than SGEAT (standard). Thus using the non-toxic samples of SGEAT (heuristic)  may not only limit the diversity of SGEAT (augmented) but also lead to higher toxicity scores, thus making it less efficient to detoxify LMs. We add more discussion in Appendix B.
> | Methods | distinct-1 | distinct-2 | distinct-3 | distinct-4 |
> |------------| -----------|------------|------------|------------|
> |SGEAT(heuristic) | 0.009 | 0.070 |  0.159 | 0.219  |
> |SGEAT(standard) | 0.039 | 0.282 |  0.615 | 0.828  |
>
> **Minor:**
>
> “SGEAT and SGEFT are used interchangeably throughout the paper including Figure 2 and Table 12 while SGEFT was not defined anywhere in the paper. Is it a typo?”
> * Thank you so much for pointing it out. It is indeed a typo. We have updated it in the revision.
>
> **Limitations:**
>
> "The main limitation of this work is the error propagation from the automatically generated training set and the Perspective API that has been used to filter the non-toxic samples.”
> * Thanks for your comment. We discussed the limitations of our work in Appendix D.4. We acknowledge that the errors can come from both self-generation and Perspective API. We note that larger models are less likely to generate unnatural languages and grammatical errors, thus mitigating the errors. We also observe that Perspective API  is not perfect, but we believe that SGEAT can also get more benefits with a more robust, unbiased, and fair hate speech detector.

---

### Official Review · Reviewer_qG1s · 2022-07-11

**Rating:** 7
**Confidence:** 3
**Soundness:** 3 good
**Presentation:** 3 good
**Contribution:** 3 good

**Summary:**

This paper explores domain-adaptive training (i.e., continued pretraining on special data to remove unwanted model behavior) with respect to the influence of model size, training corpus, and parameter efficiency. The authors further propose self-generation enabled domain-adaptive training (or "SGEAT"), where a self-generated dataset is used for detoxification. SGEAT can also be beneficially combined with a decoding-time method for detoxification.
The authors find that, SGEAT reduces model toxicity, but that model size has little effect on model toxicity. (I.e., the toxicity comes from the data.) They also show that it is more challenging to detoxify large language models.

**Questions:**

- In the paper you list your hypotheses regarding why SGEAT should work better than existing approaches. Is there a way to explore your assumptions in more detail?
- What are the limitations of your work?

**Limitations:**

I don't think there are obvious negative societal impacts of this work. The authors do not explicitly address limitations.

**Strengths And Weaknesses:**

Strengths:
- Work on detoxifying language models is important in order to safely deploy them. Thus, the topic of the paper is highly relevant.
- The fact that SGEAT works is interesting: apparently the model benefits from seeing its own (detoxified) data.
- The authors use both automatic and human evaluation in their experiments.

Weaknesses:
- Some of the findings are rather straightforward/potentially not novel, even though I don't know references from the top of my head. For example the trade-off between detoxification effectiveness and model performance is obvious (as our datasets are biased/toxic). Similar for the fact that detoxifying larger models is harder.

---

> ### Author Response · Authors · 2022-08-02
> **Response to Reviewer qG1s**
>
> Thank you so much for your review. The suggestions are really helpful to improve our paper quality. We will address your comments in the following.
>
> 1, “Some of the findings are rather straightforward/potentially not novel, even though I don't know references from the top of my head. For example the trade-off between detoxification effectiveness and model performance is obvious (as our datasets are biased/toxic). Similar for the fact that detoxifying larger models is harder.”
> * We agree that some of our findings are not surprising, but it takes nontrivial efforts to establish them with solid & extensive empirical results. Moreover, some of our findings are not straightforward. For example, it is well known that large-scale LMs are good few-short learners, which suggests that they might be easier to adapt to the nontoxic domain. However, we find that detoxifying larger models requires more effort, which suggests that larger models are more difficult to **unlearn** toxic samples in the training data.
>
> 2, “In the paper you list your hypotheses regarding why SGEAT should work better than existing approaches. Is there a way to explore your assumptions in more detail?”
>
> * Thank you for the suggestion. We hypothesize that the high efficiency is due to the fact that self-generated data are sampled from the high-density regions of the output distribution of a pre-trained LM. We leverage the PPL to verify it. If the generated corpus shows a higher PPL, it means that the corpus better captures the high-density region of the  LM. We evaluate and compare the PPL of the generated corpus of SGEAT (augmented) and OWTC by the standard 1.3B LM. The results are shown in the table below.
>
> |   | SGEAT (augmented) | OWTC |
> |------------| -----------|------------|
> | PPL of data | 5.98 | 7.93 |
>
> We find that SGEAT (augmented) demonstrates a much lower PPL than OWTC, which confirms our hypothesis that our generated corpus SGEAT (augmented) better captures the high-density regions of the LM output space. We add more discussion in the revision in Appendix C.3
>
> 3, “What are the limitations of your work?”
> * Thanks for raising the question. We discussed the limitations of work in Appendix D.4 at submission for space reason. We will move the discussion to the main text in the final manuscript. In summary, the limitations are from two aspects:
>     * First, we recognize that similar to DAPT, SGEAT also relies on Perspective API. Since Perspective  API is imperfect and is known to amplify the social bias against different demographic groups, SGEAT may also be impacted due to the use of Perspective API for filtering. However, we believe that SGEAT can also get more benefits with a more robust, unbiased, and fair hate speech detector.
>     * Second, in this paper, we mainly focus on the intrinsic quality of LMs and analyze the trade-off between toxicity and quality. While recent work demonstrates that detoxification methods may amplify social biases, we leave it as a future work to analyze the bias impact after detoxification.

---

### Official Review · Reviewer_MutJ · 2022-07-12

**Rating:** 6
**Confidence:** 3
**Soundness:** 2 fair
**Presentation:** 3 good
**Contribution:** 2 fair

**Summary:**

* This paper investigates language detoxification in multiple settings: training corpus, model size, and efficient modeling.
* Also, it proposes a new detoxification method training language models (LM) with a set of self-generated non-toxic text, which is a smaller corpus compared to OWTC.
* The experiments show that the training corpus matters the most for the detoxification, as the model size gets bigger the more effort is required (more training epoch or larger corpus) and the importance of the adaptor, which reduces the detoxification while improving the perplexity of the LM.
* Overall: I think this paper introduces variant DAPTs on different sets of training datasets and a different architecture (adapter). This paper is more toward an analytic research paper finding components which must considered for a detoxification model, which I do believe is meaningful. However, the originality of the proposing method is weak.

**Questions:**

1. Is there a reason for using the utility of average accuracy? Toxicity scores already indicate the accuracy of toxicity of generated text.
2. In section 3, regarding the pertained LM, shouldn't the models be represented as GPT-2 small, medium, and xl instead of GPT-3? GPT-3 itself is known to be a model using 175B parameters. To be more accurate, I think GPT-2 makes more sense than GPT-3.
3. Have you experimented with a similar setting with OWTC which uses the random sampled 100K from 7500K and from the sampled set utilizing the top 35% (~50K) nontoxic texts for the training?
    * I think that Figure 2 should experiment on the same dataset size to claim the model is overfitting.
4. In Table 2&10, SGEAT(heuristic) tends to have more toxic text than SGEAT(standard&augmented). The paper argues that SGEAT(heuristic) is more toxic than the other two SGEAT data diversity. How diverse are the sentences generated? Could you evaluate using distinct-1, distinct-2 & distinct-3, or another metric that indicates the diversity of generated sentences?
5. Is the validation set for the ppl evaluation derived from the non-toxic 150K OWTC? Or each training corpus
6. How is DEXPERTS with SGEAT (augmented) formed? DEXPERTS is consist of two auxiliary models, expert, and anti-expert. Did you train both two models using SGEAT (augmented)? Or did you only use the detoxified one?

**Ethics Review Area:**

["Inappropriate Potential Applications & Impact  (e.g., human rights concerns)"]

**Limitations:**

* This paper has well stated the limitation of the SGEAT.


**Strengths And Weaknesses:**

* Originality
    *  The concept of training with self-generated text for detoxification is proposed in dialogue generation [1]. The approach might not be novel to the task requiring detoxification.
* Quality
    * The purpose and direction of this paper are good. But there are claims that need to be justified (3,4).
* Clarity
    * The most of writing is clearly written. Few of them need more explanation or clarification (1,2,5,6).
* Significance
    * With a smaller refined training corpus, SGEAT(augmented) achieved better performance than the baselines. This paper conduct exhaustive experiments and analyses on large-scale models.
    * However, the prompt construction for the heuristic result is similar to an attribute conditioning model [2], which results in high toxicity scores compared to standard or augmented. The experiments on prompt construction seem unnecessary for the proposing model SGEAT.

[1] Baheti, Ashutosh, et al. "Just say no: Analyzing the stance of neural dialogue generation in offensive contexts." arXiv preprint arXiv:2108.11830 (2021).

[2] Gehman, Samuel, et al. "Realtoxicityprompts: Evaluating neural toxic degeneration in language models." arXiv preprint arXiv:2009.11462 (2020).

---

> ### Author Response · Authors · 2022-08-02
> **Response to Reviewer MutJ (Part 1)**
>
> Thank you for your detailed comments and valuable suggestions. They are really helpful to improve the quality of our paper. We will address your comments in the following.
>
> **About Originality:** “However, the originality of the proposing method is weak.” … “The concept of training with self-generated text for detoxification is proposed in dialogue generation [1]. The approach might not be novel to the task requiring detoxification.”
>
> * We thank the reviewer for pointing out this related work. We cite and discuss this work in the paper revision in Section 2. Although both works use self-generated text for detoxification, the motivation and scope are very different. In this work, we investigate the general purpose detoxification techniques for large-scale pre-trained language models, and the detoxified LM can later be used for various downstream tasks without performance degradation (e.g., QA, summarization, reasoning, and dialogue). This stringent requirement for future transfer learning has important implications for the methodology and evaluation metrics.
>     * i) In terms of methodology, our self-generation method is motivated by mitigating the exposure bias issue of autoregressive LM during domain-adaptive training, where the LM is teacher-forced and trained on non-toxic data. As a result, we perform comprehensive studies to investigate the efficient prompt strategies to better prompt the LMs to generate diverse output, in order to cover & reduce the potential toxic output in future downstream applications. In contrast, [1] uses the existing Reddit corpus to generate follow-up replies in a conversation.
>     * ii) In terms of evaluation, we need to carefully monitor the validation perplexity on a hold-out subset of the pre-training corpus, because the valid PPL is an important indicator of LM’s capability for future downstream tasks. We also test detoxified LMs on nine different downstream tasks, covering question answering, natural language understanding, and commonsense reasoning, then report the average accuracy when we apply detoxification to a pre-trained LM. In contrast, [1] focuses on the toxicity issue for a particular downstream task, where the autoregressive LM decoder has already been fine-tuned on dialog data. As a result, one need only measure the quality of the generated response.
>
> [1] Baheti, Ashutosh, et al. "Just say no: Analyzing the stance of neural dialogue generation in offensive contexts." arXiv preprint arXiv:2108.11830 (2021).
>
> **Questions:**
>
> 1, “Is there a reason for using the utility of average accuracy? Toxicity scores already indicate the accuracy of toxicity of generated text.”
> * This is also relevant to our previous response. In this work, we focus on detoxifying pre-trained LMs for general purposes. The utility of average accuracy of LM is defined as the average accuracy on 9 different downstream tasks. We use it because we want to ensure the detoxified LM can later be used  for various downstream tasks without performance degradation in contrast to the original pretrained LM. The toxicity scores indicate the levels of toxicity of generated text, and are not related to the utility average accuracy.
>
> 2, “In section 3, regarding the pertained LM, shouldn't the models be represented as GPT-2 small, medium, and xl instead of GPT-3? GPT-3 itself is known to be a model using 175B parameters. To be more accurate, I think GPT-2 makes more sense than GPT-3.”
> * Thank you for raising the question. GPT-3 Small, Medium, and XL are known terms defined in the GPT-3 paper (see Table 3.11 in [2]). Because besides model parameters, one important difference between GPT-3 and GPT-2 is that GPT-3 leverages a much larger pre-training dataset than GPT-2. All of our LMs in this work are pre-trained on a dataset of a similar size to GPT-3 in the same setting, which is why we compare it with GPT-3 XL.
>
> [2] Brown, Tom B. et al. “Language Models are Few-Shot Learners.”  (2020)

---

> ### Author Response · Authors · 2022-08-02
> **Response to Reviewer MutJ (Part 2)**
>
> **Questions: (continued)**
>
> 3, “Have you experimented with a similar setting with OWTC which uses the random sampled 100K from 7500K and from the sampled set utilizing the top 35% (~50K) nontoxic texts for the training? I think that Figure 2 should experiment on the same dataset size to claim the model is overfitting.”
> * Thank you for the valuable suggestion. We follow your suggestion and sample a subset of OWTC, which consists of 50K nontoxic documents (the same number of documents as SGEAT (augmented)). We fine-tune the 530B LM with the sampled nontoxic OWTC dataset. We evaluate the toxicity, and its corresponding PPL as in the updated Figure 2. We also list the results in the following table.
>
> | SGEAT (augmented, 50k) | 1 epoch | 3 epoch | 5 epoch |
> |------------| -----------|------------|------------|
> |Exp. Max. Toxicity | 0.491 |  0.465 | 0.440 |
> |PPL | 6.72 |   7.86 |  9.63 |
>
> | DAPT (OWTC, 50k) | 1 epoch | 3 epoch | 5 epoch |
> |------| -----------|------------|------------|
> |Exp. Max. Toxicity | 0.531 | 0.521 | 0.518 |
> |PPL | 6.438 | 6.74 | 7.17 |
>
> * We note that the curve of DAPT (OWTC, 50k) is similar to SGEAT (augmented, 50k), where the model toxicity first drops significantly (steep slope), then gradually adapts towards the nontoxic domain, and finally becomes flatter. We also observe that reducing the size of the dataset makes DAPT less efficient to detoxify 530B LM, which further confirms that large-scale LMs require more endeavors (e.g., more training data) to detoxify. We add more discussion in the revision in Section 5.
>
> 4, “In Table 2&10, SGEAT(heuristic) tends to have more toxic text than SGEAT(standard&augmented). The paper argues that SGEAT(heuristic) is more toxic than the other two SGEAT data diversity. How diverse are the sentences generated? Could you evaluate using distinct-1, distinct-2 & distinct-3, or another metric that indicates the diversity of generated sentences?”
> * Thank you for the valuable feedback. We follow your suggestion, and evaluate the diversity of SGEAT (heuristic), SGEAT (standard), and OWTC using distinct-1, distinct-2, distinct3 & distinct-4. The results are evaluated on the whole generated corpus as shown in the Table below.
>
> | Methods | distinct-1 | distinct-2 | distinct-3 | distinct-4 |
> |------------| -----------|------------|------------|------------|
> |SGEAT(heuristic) | 0.009 | 0.070 |  0.159 | 0.219  |
> |SGEAT(standard) | 0.039 | 0.282 |  0.615 | 0.828  |
> | OWTC | 0.049 | 0.336 | 0.670 | 0.854 |
>
> * We find that SGEAT (heuristic) indeed generates less diverse data than SGEAT (standard), and thus limits the effectiveness of detoxification. In contrast, the diversity of SGEAT (standard) is relatively close to the real-world corpus OWTC.  We add more discussion in the revision in Appendix B.5.
>
> 5, “Is the validation set for the ppl evaluation derived from the non-toxic 150K OWTC? Or each training corpus”
> * The validation set is a held-out subset of the pre-training corpus from 15 high-quality datasets, which does not involve data filtering in terms of toxicity. We also evaluate the model PPL on the filtered nontoxic portions of the validation set in Table 18 in Appendix C.2. We observe that 1) the PPL increases on the nontoxic subsets of the validation corpus is less than the full validation set; 2) the trend of PPL increases on nontoxic subset is similar to that on the full validation set.
>
> 6, “How is DEXPERTS with SGEAT (augmented) formed? DEXPERTS is consist of two auxiliary models, expert, and anti-expert. Did you train both two models using SGEAT (augmented)? Or did you only use the detoxified one?”
> * The expert model is fine-tuned on nontoxic 150k OWTC, which is the same as DAPT (nontoxic); while the anti-expert model is fine-tuned on the toxic portion of 150k OWTC. To conduct a controlled comparison, both SGEAT + DEXPERTS and standard DEXPERTS use these same expert and anti-expert models as their auxiliary models. The only difference is that SGEAT + DEXPERTS uses the SGEAT (augmented) model as the main generative model, while the standard DEXPERTS uses the standard 1.3B LM as the main generative model. The results demonstrate that SGEAT (augmented) model can be seamlessly combined with other decoding-time methods, even the auxiliary models are trained on different datasets. We clarify this in the revision in Section 4.3.
>
> 7, “Ethics Flag: Inappropriate Potential Applications & Impact  (e.g., human rights concerns)”
> * We also care about the well-being of human annotators. As mentioned in Line 565-567, before conducting our study, it was reviewed by our Institutional Review Board (IRB). As pointed out in Line 365-367, we carefully consider the annotators’ well-being and ensure that the average number of toxic samples (TOXICITY >= 0.5 evaluated by Perspective API) is less than or equal to 3 in each batch of 10 samples for human annotation.

---

> > ### Comment · Reviewer_MutJ · 2022-08-08
> > **Response to Author**
> >
> > 3. Thank you for experimenting additional model. I could observe from your result the relationship between dataset size and model size to make a detoxification model.
> > I think you have experimented on DAPT (OWTC, 50k) as how you collected the SGEAT (augmented, 50k), but to double check, did you randomly sample the dataset or filter out toxic set?
> >
> > 4. Thank you for adding evaluation on distinct-n. Did you use the same distinct metric as DExperts [1]? The result follows the diversity distribution similar to the OWTC dataset. But it seems to be too small. Could you show one other baseline (DAPT or DExperts)? Also, since there is a performance difference between SGEAT and SGEAT (augmented, 50k) has the best performance, I think SGEAT (augmented, 50k) also must be observed.
> >  If possible, it would be great if I could see two more distinct-n results on the baseline and SGEAT (augmented, 50k).
> >
> > 5-6. Thank you for the clarification.
> >
> > 7. I will remove the ethical flag. I missed the part. Thank you for kindly indicating human evaluation section.
> >
> > [1] https://github.com/alisawuffles/DExperts (check DExperts/scripts/evaluation/evaluate_generations.py)

---

> > > ### Author Response · Authors · 2022-08-09
> > > **We appreciate your valuable feedback**
> > >
> > > Thank you so much for the valuable follow-up feedback! We hope our response could address your major concerns.
> > >
> > > 3, “I think you have experimented on DAPT (OWTC, 50k) as how you collected the SGEAT (augmented, 50k), but to double check, did you randomly sample the dataset or filter out toxic set?”
> > >
> > > * To be more clear, we first filter out all the toxic samples and choose the top 2% (150k) nontoxic samples from the 7500k OWTC corpus to form DAPT (nontoxic OWTC, 150k). We then randomly sample 50k nontoxic samples from the top nontoxic 150k OWTC corpus to form DAPT (nontoxic OWTC, 50k) so that it has a similar data distribution as 150k OWTC but only has fewer data.
> > >
> > > 4, “Did you use the same distinct metric as DExperts [1]? The result follows the diversity distribution similar to the OWTC dataset. But it seems to be too small.”
> > >
> > > * Thanks for pointing it out. We used the same evaluation metrics as DExperts. However, we note that DExperts is evaluating the **per prompt generations** given the RealToxicityPrompts benchmark, where each generation only consists of 20 tokens. In contrast, we evaluate the **whole training corpus** (OWTC or self-generated corpus), where the whole document consists of around 3 million tokens. So this is the reason why our diversity numbers look too small, as the diversity metrics will be the number of distinct tokens divided by the number of total tokens.
> > >
> > > * We also follow your suggestion, and evaluate the diversity of **per prompt generations** as in the DExperts paper.
> > >
> > > “Could you show one other baseline (DAPT or DExperts)? Also, since there is a performance difference between SGEAT and SGEAT (augmented, 50k) has the best performance, I think SGEAT (augmented, 50k) also must be observed. If possible, it would be great if I could see two more distinct-n results on the baseline and SGEAT (augmented, 50k).”
> > >
> > > * Thanks for the valuable suggestion. We follow the setup of DExperts and evaluate the generation diversity of 1.3B models given the RealToxicityPrompts benchmark. Specifically, generation diversity is measured using the average number of distinct n-grams, normalized
> > > by the length of text, for each prompt. We report Distinct-1, Distinct-2, and Distinct-3 scores for distinct uni-, bi-, and trigrams, respectively.
> > > | Model | distinct-1 | distinct-2 | distinct-3 |
> > > |------------| -----------|------------|------------|
> > > |Standard 1.3B | 0.5662 | 0.9076 |  0.9128 |
> > > |DAPT (nontoxic OWTC, 150k) | 0.5615 | 0.8887 | 0.9065 |
> > > |SGEAT(augmented, 50k) | 0.5546 | 0.8923 |  0.9087 |
> > > |SGEAT(standard, 50k) | 0.5506 | 0.9002 |  0.9110 |
> > > |SGEAT(heuristic, 50k) | 0.5407 | 0.8884 |  0.9090 |
> > >
> > > * We find that after detoxification, the generation diversity of SGEAT (augmented), SGEAT (standard), and the baseline DAPT drops mildly, while the diversity of SGEAT (heuristic) indeed drops more. This again suggests that SGEAT (heuristic) is less effective to detoxify LM due to the limited data diversity.

---

> > > > ### Author Response · Authors · 2022-08-09
> > > > **Follow-up Response to Reviewer MutJ**
> > > >
> > > > Dear Reviewer MutJ,
> > > >
> > > > As the end of the discussion is approaching, we really appreciate it if you can consider our follow-up responses to your questions when evaluating our work. We are very thankful for your comments and suggestions that helped improve our paper and we have incorporated them in our revision. We made an earnest effort to address your questions, and would greatly appreciate it if you would consider raising your score in light of our response. Please let us know if you have any final questions that we can address. Thanks!
> > > >
> > > > Best regards,
> > > > Authors

---

> > > > > ### Comment · Reviewer_MutJ · 2022-08-09
> > > > > **Response to Author 2**
> > > > >
> > > > > 3.  Thank you for your detailed explanation. I understand how the OWTC dataset is formed for nontoxic DAPT. My concern was that to truly evaluate whether the self-generated sentences are the most influential factor in SGEAT, I felt an ablation experiment with/without “Data Filtering” is necessary. I was asking if you have conducted an experiment on a nontoxic OWTC dataset filtered the same way as SGEAT. By constraining two models with similar toxicity distributed datasets but from different sources, I thought that experiment would make your model’s contribution more apparent. I guess it was not clearly delivered from the initial review.
> > > > >
> > > > > 4. Thanks for computing the “per prompt generations” diversity. It resolves my concern about the generation quality. I was concerned with the high percentage of word repetition in the generated text.
> > > > >
> > > > >
> > > > > Comment: In Appendix C3, "If the generated corpus shows a higher PPL…”, this statement needs to be changed to a lower PPL. The smaller PPL values indicate LM has learned similar word distributions to the comparison target.
> > > > >
> > > > >
> > > > > I agree with you that large-scale research on detoxification and domain adaptation beyond detoxification is important that must be researched. However, my concern is still unresolved in the case of what if the nontoxic OWTC 50K dataset, filtered the same way as SGEAT, results in a similar performance as the SGEAT. Without the strong confidence that the unfiltered self-generated corpus itself results in better performance in detoxification as well as the PPL, my score remains the same.

---

> > > > > > ### Author Response · Authors · 2022-08-09
> > > > > > **Important clarifications**
> > > > > >
> > > > > > Thank you for raising the question.
> > > > > >
> > > > > > > "In Appendix C3, "If the generated corpus shows a higher PPL…”, this statement needs to be changed to a lower PPL."
> > > > > >
> > > > > > - Thanks for pointing it out. We are sorry for the typo. We have fixed it and updated our draft.
> > > > > >
> > > > > > > 3, "I was asking if you have conducted an experiment on a nontoxic OWTC dataset filtered the same way as SGEAT. By constraining two models with similar toxicity distributed datasets but from different sources, I thought that experiment would make your model’s contribution more apparent."
> > > > > >
> > > > > > - Actually, nontoxic DAPT applies much more aggressive filtering than SGEAT. Specifically, DAPT filters 98% of the OWTC corpus and maintains the top 2% nontoxic OWTC for fine-tuning. In Table 12 (rightmost column), its average toxicity after filtering is 0.01 +- 0.01. In contrast, SGEAT only filters 50% of the generated corpus, and maintains the top 50% nontoxic corpus for fine-tuning. In Table 12, its average toxicity after filtering is 0.03 +- 0.02.
> > > > > >
> > > > > > - When performing domain-adaptive training or fine-tuning, recent studies [1] show that if the training/fine-tuning set is filtered more aggressively and has lower data toxicity, then the fine-tuned LMs have lower toxicity in generations. However, we have shown that, even SGEAT’s fine-tuning set has higher toxicity scores than nontoxic DAPT's (0.03 +- 0.02 vs. 0.01 +- 0.01), SGEAT can still achieve better detoxification performance than nontoxic DAPT as shown in our main Table 2. One major reason is that the self-generated fine-tuning corpus can mitigate the exposure bias issue, thus it is more effective to reduce the toxicity of LMs.
> > > > > >
> > > > > > - As a result, the nontoxic OWTC 50K dataset, if it is filtered the same way as SGEAT (i.e., both of them have the same toxicity dataset distribution: 0.03 +- 0.02), the performance will only be worse than the currently provided nontoxic OWTC 50K result, which is filtered with lower data toxicity (i.e., 0.01 +- 0.01). We thought you were asking for the result this setting (i.e., toxicity dataset distribution: 0.01 +- 0.01), because it actually favors nontoxic DAPT method.
> > > > > >
> > > > > > [1] Welbl, Johannes et al. “Challenges in Detoxifying Language Models.” Findings in EMNLP (2021)

---

> > > > > > ### Author Response · Authors · 2022-08-10
> > > > > > **Experiment result on a nontoxic OWTC dataset filtered the same way as SGEAT**
> > > > > >
> > > > > > Dear Reviewer,
> > > > > >
> > > > > > This is a continued reply with the empirical evidence that supports our claim in the previous clarification response:
> > > > > >
> > > > > > > The nontoxic OWTC 50K dataset, if it is filtered the same way as SGEAT (i.e., both of them have the same toxicity dataset distribution: 0.03 +- 0.02), the performance will only be worse than the currently provided nontoxic OWTC 50K result, which is filtered with lower data toxicity (i.e., 0.01 +- 0.01)
> > > > > >
> > > > > >
> > > > > > **The new nontoxic OWTC 50K dataset is filtered the same way as SGEAT, and it has a similar toxicity distribution as SGEAT, i.e., its average toxicity is 0.03 +- 0.02.**  We refer to DAPT using this training set as DAPT (OWTC-0.03, 50k), and the previous one with 0.01 average toxicity as DAPT (OWTC-0.01, 150k).  We refer to SGEAT as SGEAT (augmented-0.03), because its average dataset toxicity is 0.03. The fine-tuning results for 3 epochs on 1.3B LM is shown below.
> > > > > >
> > > > > > |  | SGEAT (augmented-0.03) | DAPT (OWTC-0.03, 50k)  | DAPT (OWTC-0.01, 150k) |
> > > > > > |------------| -----------|------------|------------|
> > > > > > |Exp. Max. Toxicity | 0.43 |  0.51 | 0.47 |
> > > > > > |Toxicity Prob. | 37% | 47% | 43% |
> > > > > > |PPL | 11.19 |   10.43  | 10.40 |
> > > > > >
> > > > > > The results show that with higher data toxicity in the new nontoxic OWTC 50k dataset, DAPT (OWTC-0.03, 50k) exhibits higher toxicity in generations than DAPT (OWTC-0.01, 150k) and SGEAT (augmented). This confirms that given the similar toxicity distribution (i.e., 0.03 +- 0.02) and the same amount of data (i.e., 50k), SGEAT (augmented) can achieve much better detoxification effectiveness.
> > > > > >
> > > > > > Because the author-reviewer discussion is closing soon, we don’t have enough time to fine-tune the 530B LM as we did in the rebuttal. So, we conduct the same experiments on 1.3B LM. The same conclusion should be carried to the 530B model. We will provide the 530B result in our final manuscript.
> > > > > >
> > > > > > We hope the above results could resolve your last concern:  "**what if the nontoxic OWTC 50K dataset, filtered the same way as SGEAT, results in a similar performance as the SGEAT**".
> > > > > >
> > > > > > Many thanks,
> > > > > >
> > > > > > The Authors

---

> > > > > > > ### Comment · Reviewer_MutJ · 2022-08-10
> > > > > > > **Thank you for addressing my last concern**
> > > > > > >
> > > > > > > You did it! I will definitely raise my score.

---

> ### Author Response · Authors · 2022-08-07
> **Thank you for your reviews**
>
> Thank you for your insightful comments and valuable suggestions. They are really helpful to improve the quality of our paper.
>
> We follow your suggestions and revise the papers accordingly. We are wondering if our responses have resolved your concerns. Please let us know if you have any questions. Thank you!

---

### Review · Ethics_Reviewer_1De5 · 2022-08-03

**Recommendation:** Yes, see above.

**Ethical Issues:**

Yes

**Ethics Review:**

This paper was flagged for ethical review by Reviewer MutJ, but said reviewer did not provide any explanation for the flag unless I missed that explanation (apologies if I did).

Having said that, there is an area or two I think the authors need to do more on:

1. "Submissions to NeurIPS are expected to include a discussion about potential negative societal impacts of the proposed research artifact or application." In the checklist of this paper, the authors indicate that "Our work focus _[sic]_ reducing the toxicity of LMs." Beyond that point, I struggle to find a dedicated discussion of the potential negative societal impacts. More of a discussion regarding the potential negative societal impacts occurs in the `Discussion and Case Studies` section, especially at `D.3 Bias against Marginalized Groups` and `D.4 Limitation of SGEAT`. In `D.4`, the authors note that "Since Perspective API is imperfect and is known to amplify the social bias against different demographic groups, SGEAT may also impacted due to the use of Perspective API for filtering. However, we believe that SGEAT can also get more benefits with a more robust, unbiased, and fair hate speech detector." I think the authors could do more to elaborate on this point.
#
2. "Second, in this paper, we mainly focus on the intrinsic quality of LMs and analyze the trade-off between toxicity and quality. While recent work demonstrates that detoxification methods may amplify social biases, we leave it as a future work to analyze the bias impact after detoxification." I think it's fine to point to future work as a limitation, but a discussion here is likely warranted. To the extent recent work demonstrates that detoxification methods may actually _amplify_ social biases, and to the extent that the experiments at issue here show that the training corpus matters the most for the detoxification, what does that say about detoxification approaches and/or LMs? To my eye, it does seem a bit as if the paper is punting some of those considerations for future work. Again, this isn't to say one has to forecast the results of that work, but I do think engaging with the substance a bit would be helpful.
#
3. Finally, I don't know how this is traditionally approached, but I did find it notable that the final example in `Table 24` for Model: SGEAT (augmented) Prompt: "I would to talk to you about your comments about my being a racist," contains a racial slur. I would consider redacting that word such that the final five characters are *****.

---

### Review · Ethics_Reviewer_dg67 · 2022-08-04

**Recommendation:**

While the authors authors acknowledge the potential tradeoffs with regards to toxicity and bias, they note that the scope of future research should focus on mitigating social biases on pre-trained models when an equally or more responsive recommendation for future research would also be to consider the potential harms of mitigating toxicity at the expense of amplifying racial bias (and acknowledging that these phenomena may are not necessarily distinct).

**Ethical Issues:**

Yes

**Ethics Review:**

Datasets, especially machine generated ones, to evaluate and mitigate toxicity are likely to implicate social biases and the use of existing models to generate new training or evaluation data risks propagating bias that may be reflected in those models.

Exposing human annotators to too much toxic content could have an effect on their mental health.

---

### Author Response · Authors · 2022-08-02
**General Response**

We thank all the reviewers for their valuable time and detailed comments. They are really helpful to improve our paper. We have addressed the comments and included more experimental results by following the suggestions from reviewers.

Specifically, we made the following revisions:
1. (Reviewer MutJ) We added more discussion about [1] in Section 2 Related Work.
2. (Reviewer MutJ) We updated Figure 2 and added a new curve DAPT (OWTC, 50k), which is experimented on the OWTC dataset of the same size as SGEAT to be more fair.
3. (Reviewer MutJ, Pnxe) We added a new section B.5 discussing the data diversity evaluation.
4. (Reviewer MutJ) We added more details of DExperts in Section 4.3.
6. (Reviewer qG1s) We added more discussion and experimental results about why self-generation is more data efficient than pre-training corpus in Appendix C.3.
7. (Reviewer Pnxe) We revised the typos in Figure 2 and Table 14.
8. (Reviewer Pnxe) We added more discussion on transferring self-generated from larger LMs to smaller LMs in Appendix C.4.
9. (Reviewer Pnxe) We updated our claims about the trade-off between toxicity and perplexity in Section 5.
10. (Reviewer zpdX) We added more discussion on mixing self-generated data with OWTC in Appendix C.5.
11. (Reviewer zpdX) We added our summaries on the guidance of how to choose the adaptation methods in Appendix D.1.

All of our revisions are updated in OpenReview now and highlighted in blue. Thank you!

[1] Baheti, Ashutosh, et al. "Just say no: Analyzing the stance of neural dialogue generation in offensive contexts." arXiv preprint arXiv:2108.11830 (2021).

---

### Meta-Review · Area_Chair_JRar · 2022-08-21

**Recommendation:** Accept
**Confidence:** Less certain

**Metareview:**

This paper explores domain-adaptive training (i.e., continued pretraining on special data to remove unwanted model behavior) with respect to the influence of model size, training corpus, and parameter efficiency. All technical reviewers lean toward accepting this paper. However, there are also some ethics concerns.

**Award:**

No

---

### Decision · Program_Chairs · 2022-09-14

Accept